# HighDimMixedModels.jl: Robust high-dimensional mixed-effects models across omics data

**Evan Gorstein**[1,2], **Rosa Aghdam**[1], **Claudia Solís-Lemus**[1,3]*

1 Wisconsin Institute for Discovery, University of Wisconsin-Madison, Madison, Wisconsin, United States of America, 2 Department of Statistics, University of Wisconsin-Madison, Madison, Wisconsin, United States of America, 3 Department of Plant Pathology, University of Wisconsin-Madison, Madison, Wisconsin, United States of America

* solislemus@wisc.edu

## Abstract

High-dimensional mixed-effects models are an increasingly important form of regression in which the number of covariates rivals or exceeds the number of samples, which are collected in groups or clusters. The penalized likelihood approach to fitting these models relies on a coordinate descent algorithm that lacks guarantees of convergence to a global optimum. Here, we empirically study the behavior of this algorithm on simulated and real examples of three types of data that are common in modern biology: transcriptome, genome-wide association, and microbiome data. Our simulations provide new insights into the algorithm's behavior in these settings, and, comparing the performance of two popular penalties, we demonstrate that the smoothly clipped absolute deviation (SCAD) penalty consistently outperforms the least absolute shrinkage and selection operator (LASSO) penalty in terms of both variable selection and estimation accuracy across omics data. To empower researchers in biology and other fields to fit models with the SCAD penalty, we implement the algorithm in a Julia package, `HighDimMixedModels.jl`.

## Author summary

High-dimensional, clustered data are increasingly common in modern omics. In our study, we focus on the penalized likelihood approach to fitting mixed-effects models to these data, employing a coordinate descent (CD) algorithm to minimize the objective function. Although CD is a common optimization scheme, its convergence in this setting lacks guarantees, prompting our empirical investigation of its behavior when applied to transcriptome, genome-wide association, and microbiome datasets. We evaluate the model and algorithm's performance on simulations of these studies and subsequently apply it to real examples of each. To help facilitate the practical application of these models and further research, we have implemented the algorithm in an open-source Julia package, `HighDimMixedModels.jl`. This package provides implementations of both the least absolute shrinkage and selection operator (LASSO) and the smoothly clipped

**Data Availability Statement:** The high dimensional mixed-effects model is open source and publicly available as a Julia package HighDimMixedModels.jl at https://github.com/solislemuslab/HighDimMixedModels.jl. All simulation and real

data analyses scripts are available in github.com/
evangorstein/hdmmExperiments. We provide a
table in this GitHub repository summarizing many
other existing methods and software for fitting
penalized mixed-effects models to biological data:
https://bit.ly/All-paper-for-mixed-effect-models.

**Funding:** This work was supported by the
Department of Energy [DE-SC0021016 to CSL] and
by the National Science Foundation [DEB-2144367
to CSL]. The funders did not play any role in the
study design, data collection and analysis, decision
to publish, or preparation of the manuscript.

**Competing interests:** The authors have declared
that no competing interests exist.

absolute deviation (SCAD) penalty, and having tested its performance on various omics
data sets, we hope that it offers a user-friendly solution for researchers in biology.

## 1 Introduction

High-dimensional regressions, in which the number of variables matches or exceeds the number of samples, are the rule, rather than the exception, in modern omics. In the field of transcriptomics, for example, RNA-Seq technology allows for simultaneous measurement of expression levels of thousands of genes [1, 2]; in population genetics, genome-wide association studies include hundreds of thousands, if not millions, of single nucleotide polymorphisms (SNPs) [3, 4]; and in metagenomics, microbiome studies measure the abundance of hundreds of bacterial taxa in perhaps only a few dozen samples from the environment [5, 6]. In order to make the analyses of these data sets tractable, it is typically assumed that only a small fraction of the transcripts, SNPs, or taxa have a non-negligible impact on a response of interest, often related to human, soil, or plant health. Identifying these predictive features from the large pool of measured markers thus becomes a critical task [7].

Experimental conditions and constraints often result in omics data which, in addition to being high-dimensional, are clustered or grouped. For example, samples of a microbiome might be collected at different locations, with samples obtained from the same site more similar to each other. Alternatively, in longitudinal studies, repeated samples are taken from different individuals over time, and observations from the same individual are assumed to be correlated [8]. In a regression context, a common and flexible framework for handling grouped data of this sort is the so-called *mixed-effects model* [9]. In a mixed-effects model, in addition to estimating "fixed effects" of the measured covariates, each group present in the data is also assigned an unobserved "random effect" drawn from a common distribution. The sharing of the same random effect among units belonging to the same group induces correlation between these units.

Unfortunately, standard maximum and restricted maximum likelihood estimation of the parameters of the mixed-effects model deteriorates in the high-dimensional setting. In the case where the number of features exceeds the total sample size, there are no degrees of freedom to form the restricted likelihood, and maximum likelihood estimation entails interpolating the data with the fixed effects and sending the error variance to 0. One common approach to estimating the parameters of a high-dimensional regression model without random effects is by forming and maximizing a penalized likelihood [10]. Penalizing, for example, the $\ell_1$ norm of the regression coefficient vector in an estimation method known as the least absolute shrinkage and selection operator (LASSO) has proven to be an extremely popular approach due to (1) the ease of the corresponding numerical optimization problem and (2) the sparsity of its solutions, consequences of the geometry of the $\ell_1$ norm and its convexity [11]. The properties of the LASSO as an estimator and as a feature selector have been extensively studied and have inspired a literature that generalizes the LASSO and other sparsity-inducing regularization techniques to new domains [12].

The extension of the penalized likelihood framework to the mixed-effects model has been studied in a number of articles [13–15]. In an initial paper [13], Schelldorfer, Bühlmann, and van de Geer studied the convergence rate of the global maximizer of the penalized likelihood with an $\ell_1$ penalty and proposed a coordinate descent (CD) algorithm [16] to arrive at a local minimum of this objective function. Due to the non-convexity of the problem, such a solution does not necessarily coincide with the global optimum to which their theoretical results apply.

Ghosh and Thorensen [14] subsequently studied statistical properties of maximum penalized likelihood estimators for the high-dimensional mixed-effects model under general, non-convex penalties, including the smoothly clipped absolute deviation (SCAD) penalty, and they similarly provide a CD algorithm for implementation. However, their algorithm minimizes an adaptively modified objective function at every step, and thus the output of their algorithm is not even a local minimum of the original penalized likelihood.

Both [13] and [14] carry out simulations to study the performance of their estimation procedure under a single well-behaved design structure with normally distributed entries. In omics studies, however, design matrices often have a distinct, non-normal structure. In this paper, we test the robustness of the CD algorithm for fitting high-dimensional mixed-effects models by evaluating its statistical performance on data simulated to resemble those from the three omics studies mentioned at the outset: gene expression analyses, genome-wide association studies (GWAS), and studies of the microbiome. To facilitate this simulation study, we implemented the CD algorithm for fitting high-dimensional mixed-effects models (High-DimMM) in the Julia package `HighDimMixedModels.jl`, available at https://github.com/solislemuslab/HighDimMixedModels.jl. Our software mimics the code in the R packages `lmmlasso` and `splmm` ([17, 18]), but includes the SCAD as the default penalty, which is not available in `lmmlasso` and whose implementation in `splmm` incorrectly updates the penalized fixed effects in the CD algorithm. In addition, it corrects an error in the original code published in [14] that prevented zeroed coefficients from being further updated over the course of the algorithm.

In addition to our own, there are a myriad of methods, models, and software implementations for analyzing high-dimensional, clustered data, many of which are tailored to the analysis of a specific type of omics data. We have included a table summarizing many of these methods and software implementations at https://bit.ly/All-paper-for-mixed-effect-models [14, 15, 17–46]. Our software distinguishes itself in its speed, which is obtained through a Julia implementation of the CD algorithm, and in its ability to work robustly across high-dimensional omics data sets, as illustrated in the simulations in this paper.

## 1.1 Characteristics of targeted Omics data

Gene expression data are crucial for understanding the functional elements of the genome and the molecular mechanisms underlying various biological processes, as well as for disease diagnosis and drug development [48, 49]. These data are generated through technologies such as microarrays and RNA-Seq, which involve converting RNA molecules into complementary DNA (cDNA) and then sequencing these cDNA fragments to quantify RNA levels [47]. A key challenge associated with gene expression data is their high dimensionality: typically, thousands of genes are measured in relatively few samples [50]. In addition, gene expression data often require a normalization strategy to make the data more biologically meaningful across samples [51]. In our simulations, we assume that expression profiles can be made to follow a multivariate normal distribution by applying such a strategy, and we introduce correlations between genes in this distribution to study how this impacts algorithmic behavior and estimation performance.

In genome-wide association studies (GWAS), design matrices contain the genotypes of each individual in the study at a large collection of single nucleotide polymorphisms (SNP) across the genome. When, for a given locus in a diploid organism, there are only two distinct alleles, the individual's genotype is often represented with either a 0 (homozygous for the minor allele), 1 (heterozygous), or 2 (homozygous for the major allele); that is, the genotype is represented by the count of minor alleles, which implicitly encodes the assumption of an

additive genetic effect [52]. To avoid the computational demands of a multiple regression approach, GWAS is often done simply by testing the effect of each SNP on the phenotype in isolation of all other variants [53]. However, examining all SNPs simultaneously can improve the power to detect small effect sizes of individual SNPs, hence the need for methods to deal with the high dimensionality of GWAS data [54]. The error correlation structure for GWAS regression models is generally based on the known or estimated population substructure of the individuals included in the study [55]. In our modeling of GWAS data, we are assuming a known and very simple structure in which the individuals in the study are partitioned into discrete populations and which we account for by including random effects associated with each of these populations.

The final omics study simulated and tested in this article is that of the microbiome. The compositions of the microbial community in a collection of samples from a microbiome of interest take the form of a matrix of read counts, with each row representing a sample and each column representing a microbial taxa, often referred to as an operational taxonomic unit (OTU) [56]. However, due to differences in DNA yielding material between samples and inherent limitations in sequencing technology, counts only provide information about the relative abundance of the various OTUs in a given sample [57]. The OTU count matrix is also often extremely sparse and very right skewed [58]. In this work, we take into account the compositional nature of the data by assuming a linear log-contrast regression model for the response of interest [59]. In other words, we assume an underlying design matrix obtained as a log-ratio transformation of the original count matrix. In our simulations, we generate the response according to Eq 1 using this transformed data as our design matrix, and in both our simulations and our real data analysis, we fit models with this transformed data as the assumed design.

The remainder of the paper is structured as follows. In Section 2, we first review the high-dimensional mixed-effects model from [13] and [14], as well as the CD algorithm proposed in these sources for fitting the model with the LASSO and SCAD penalties. In Section 3, we detail the results of fitting high-dimensional mixed-effects models with CD to the simulated data and to the real data sets from each study type. We comment on the variation in the procedure's ability to recover the impactful markers and accurately estimate their effects across settings such as dimensionality, design matrix structure, and random-effects structures. We conclude with a discussion of the main takeaways from our investigation in Section 4.

## 2 Methods

### 2.1 Penalized likelihood of mixed-effects model

Let $g$ denote the number of clusters in our data. In this paper, we consider the linear mixed-effects model, where the vector of responses $y_i$ in each cluster $i = 1, \ldots, g$ is generated according to

$$y_i = X_i\beta + Z_ib_i + \epsilon_i. \tag{1}$$

We assume that we are performing variable selection of the fixed effects rather than the random effects so that $\beta \in \mathbb{R}^p$ is a high-dimensional vector of fixed effects, and $b_1, b_2, \ldots, b_g \overset{\text{iid}}{\sim} \mathcal{N}_q(0, \Psi_\theta)$ are low-dimensional vectors of random effects ($q \ll p$). Letting $n_i$ indicate the number of observations in the group $i$, $X_i \in \mathbb{R}^{n_i \times p}$ and $Z_i \in \mathbb{R}^{n_i \times q}$ are cluster-specific design matrices, with the latter corresponding to the random effects in the model, so that the variables represented as columns of $Z_i$ are taken to vary between groups in their effects on the response. These variables are typically a subset of the columns in $X_i$, so that although the

random effects are drawn from a mean zero distribution, the average effect across the clusters may be estimated non-zero as a fixed effect. Finally, for each cluster $i$, $\epsilon_i \sim \mathcal{N}_{n_i}(0, \sigma^2 I_{n_i})$ is a vector containing the error terms for each response in the cluster. The $\epsilon_i$ are assumed to be mutually independent of each other and of the random effects.

The covariance matrix $\Psi_\theta$ is a symmetric, positive semidefinite matrix that is parameterized by $\theta \in \mathbb{R}^{q^*}$. We consider three distinct models in which $\Psi_\theta$ is, in turn, a scalar ($q^* = 1$), diagonal ($q^* = q$), and arbitrary symmetric positive semidefinite ($q^* = q(q + 1)/2$) matrix. In each case, $\theta$ represents the lower-triangular Cholesky factor of $\Psi_\theta$, which can be optimized without any constraints, although for identifiably, we constrain its diagonal elements to be nonnegative.

The marginal distribution of $y_i$ in this model is $\mathcal{N}(X_i\beta, V_i(\theta, \sigma^2))$, where $V_i(\theta, \sigma^2) = Z_i\Psi_\theta Z_i^\mathsf{T} + \sigma^2 I_{n_i}$, and the log-likelihood of the full set of parameters $\phi := (\beta, \eta) := (\beta, \theta, \sigma^2)$ is thus

$$\ell(\phi) = -\frac{1}{2}\left[ N\log(2\pi) + \log(\det(V)) + (y - X\beta)^\mathsf{T} V^{-1}(y - X\beta) \right],$$

where $y \in \mathbb{R}^N$ and $X \in \mathbb{R}^{N \times p}$ are obtained by vertical stacking, $V \in \mathbb{R}^{N \times N}$ by diagonal stacking, and $N$ denotes the total sample size obtained by summing the cluster sizes, $N := \sum_{i=1}^g n_i$. We define the maximum penalized likelihood estimators to be the minimizers of the loss function

$$Q_\lambda(\phi) = -\ell(\phi) + \sum_{j=1}^p P_\lambda(|\beta_j|), \tag{2}$$

where $\lambda$ is a hyperparameter that governs the severity of penalization and must be selected. In the initial study of this model, an $\ell_1$ penalty $P_\lambda(|\beta_j|) = \lambda|\beta_j|$, was proposed [13]. As an alternative, the SCAD penalty, defined through its derivative by

$$P_\lambda'(|\beta_j|) = \lambda\left\{ I(|\beta_j| \leq \lambda) + \frac{\max(0, a\lambda - |\beta_j|)}{(a - 1)\lambda} I(|\beta_j| > \lambda) \right\},$$

was proposed for penalized regression originally in the classic low-dimensional model [60], subsequently in the high-dimensional regime [61], and finally for high-dimensional mixed-effects models in [14]. Here, $a$ is an additional tunable parameter that controls how quickly the penalty decreases, typically set to 3.7, as recommended in [60].

## 2.2 Coordinate descent algorithm

A CD algorithm for fitting high-dimensional mixed-effects model was originally proposed in [13] and is implemented in our package (Algorithm 1). At each iteration of this algorithm, we first cycle through the indices $1, 2, \ldots p$, updating each component to minimize either the original objective function $Q_\lambda$ (under the LASSO penalty) or an adaptive rescaling of it (under the SCAD penalty) with all other components fixed at their current values. For details on the analytical solution of these updates and the purpose of adaptive rescaling under the SCAD penalty, see S1 Appendix. After updating the fixed effects, we cycle through the indices $p + 1, \ldots, q^* + 1$, individually updating each variance-covariance parameter to minimize the negative log-likelihood (the only part of the objective function that depends on these parameters) while all other components are kept fixed. The iteration continues until the parameters converge or the program reaches a prespecified maximum number of iterations.

Updating the full fixed-effect vector $\beta$ at every iteration can be computationally expensive for high-dimensional problems, so following [13], we adopt an "active set" strategy: instead of

updating every component of $\beta$ at each iteration, we update only those that are currently nonzero, with updates to the full set of components made only every $D$ ($D = 5$ in our simulations) iterations. In addition, we update the full vector any time the convergence criterion is satisfied after an update of only the active set. Only when the convergence criterion is satisfied after a full update do we terminate the algorithm.

**Algorithm 1:** Coordinate descent algorithm for estimating parameters of a high-dimensional mixed-effects model

**Input:** Data `X`, `Z`, `y`, set of indices nonpen of non-penalized components of $\beta$, penalty function $P_\lambda$ (either LASSO or SCAD), initial parameter values $\beta^0$, $\eta^0$, full update frequency `D`

**// Initialize iterates**
1 $\beta \leftarrow \beta^0$; $\eta \leftarrow \eta^0$
2 **for** `k = 1, 2, ...` **do**
 **// Update fixed effect parameters**
3  **if** `k` mod `D` = 0 **then** $\mathcal{J} \leftarrow \{1, 2, \ldots, p\}$ **else** $\mathcal{J} \leftarrow \{j \mid \beta_j \neq 0\}$
4  **for** $j \in \mathcal{J}$ **do**
  **// Calculate first and second order partial derivates of the negative log-likelihood with respect to $\beta_j$**
5    $g \leftarrow -(y - X\beta)^T V(\eta)^{-1} x_j$
6    $h \leftarrow x_j^T V(\eta)^{-1} x_j > 0$
  **// Adaptively rescale penalty function if using SCAD**
7    $\tilde{P}_\lambda(|\gamma|) \overset{\text{def}}{=} \begin{cases} 0, & j \in \text{nonpen} \\ P_\lambda(|\gamma|), & j \notin \text{nonpen and penalty is LASSO} \\ P_\lambda(|h\gamma|), & j \notin \text{nonpen and penalty is SCAD} \end{cases}$
  **// Solve (analytically) univariate minimization problem and update**
8    $d^* \leftarrow \arg\min_d gd + \frac{1}{2}hd^2 + \tilde{P}_\lambda(|\beta_j + d|)$
9    $\beta_j \leftarrow \beta_j + d^*$
10   **end**
  **// Update random effect parameters**
11   **for** `j = 1, ..., `$q^* + 1$ **do**
12    $\eta_j \leftarrow \arg\min_{\eta_j} -\ell(\beta, \eta_1, \ldots, \eta_j, \ldots, \eta_{q^*+1})$
13   **end**
14   Return $\phi = (\beta, \eta)$ if convergence criterion satisfied.
**15 end**

Note that when we run this algorithm on a problem with $p > N$, we may converge to an interpolating solution, i.e. a $\hat{\beta}$ satisfying $y = X\hat{\beta}$. At this vector of fixed effects, the objective function is unbounded from below, tending to negative infinity with $\log(\det(V))$ as $\sigma^2$ goes to 0. Thus, in performing the CD algorithm, we hope to avoid this region of the optimization landscape and instead converge to a local minimum $\hat{\phi} = (\hat{\beta}, \hat{\eta})$ of $Q_\lambda$ such that $\hat{\beta}$ is sparse, that is, $\#\{j \mid \hat{\beta}_j \neq 0\} \ll p, N$. As we comment below when discussing the results of our simulations, whether the algorithm manages to avoid converging to this solution depends on the size of $\lambda$. If $\lambda$ is too small, we will converge to the interpolating solution, where the threshold for smallness is data dependent and can only be discovered by attempting to run the algorithm for different $\lambda$s.

Running Algorithm 1 constitutes a single fit of our model for a given value of the regularization hyperparameter $\lambda$. As in [13, 14], we search over a grid of $\lambda$s, fitting a model with each one, and selecting as our final model the fit that minimizes the Bayesian information criterion (BIC), $BIC_\lambda := -2\ell(\hat{\beta}, \hat{\theta}, \hat{\sigma}^2) + \hat{df}_\lambda \log N$, where the estimated degrees of freedom $\hat{df}_\lambda := |\{1 \leq j \leq p \mid \beta_j \neq 0\}| + q^*$ is the number of parameters estimated non-zero. As a lower bound for the grid of $\lambda$s, one can use a value of $\lambda$ that causes Algorithm 1 to converge to an

interpolating solution, and for an upper bound, one can use a value of $\lambda$ that causes the algorithm to converge to an estimate in which all penalized fixed effects are set to zero.

The theoretical statistical guarantees that are proven for the penalized maximum likelihood estimator with a LASSO penalty in [13] apply to the *global* minimum of $Q_\lambda$ within a restricted parameter space explicitly defined to avoid the interpolating solution. In practice, however, the output of Algorithm 1 with the LASSO penalty and when we avoid the interpolating solution is only guaranteed to be a *local* minimum, even within this restricted parameter space. Moreover, when we run Algorithm 1 with the SCAD penalty, we are not even attempting to arrive at a local minimum of the original objective function of $Q_\lambda$, as we are adaptively rescaling the objective function at each iteration, so the theoretical results in [14] do not apply. In conclusion, the theoretical results found in these papers do not directly apply to the output of the algorithm, and in this article, we study this output and the overall behavior of the algorithm when applied to omics data sets that are common in modern biology.

## 2.3 Simulated omics data

We simulated three types of omics data sets—transcriptome profiles, GWAS data, and 16S microbiome data—in order to empirically study the performance of the CD algorithm in fitting high dimensional mixed-effects models across these biologically meaningful scenarios. In each, the response was generated according to Eq (1), but the form of the design matrices $X_i$ and $Z_i$ varied considerably across the three studies so as to mimic their real-world counterparts. Within each study, we additionally varied aspects of the data such as dimensionality, sample size, and random effect structure to investigate their effect on performance. In the following sections, we detail the settings that we entertained within each of the three study types.

**2.3.1 Gene expression simulations.** To simulate gene expression design matrices, we draw a sample of $N$ i.i.d. random vectors of length $p - 1$ from a multivariate normal distribution. Viewing these vectors as (normalized) profiles of the transcritomes of the collected samples, they become the rows of our design matrix $X$; meanwhile, the first $q - 1$ components of these vectors become the rows of the matrices $Z_i$ and thus receive random effect ($p - 1$ and $q - 1$ because the final columns are set as constants for estimating fixed and random intercepts, respectively). The multivariate normal distribution from which the rows of $X$ are drawn has mean zero and a first order auto-regressive covariance matrix: $Cov(X_{i,j}, X_{i,j'}) = \rho^{|j-j'|}$ for some choice of $\rho \geq 0$. Thus, for non-zero $\rho$, columns of the design matrix that are closer to each other in index are more correlated. For the dimensionality $p$ of the problem, we consider sizes of 500 and 1000, typical of the number of genes in a small scale transcriptomics study. We set the sample size to be $N = 180$ when $p = 500$ and $N = 250$ when $p = 1000$, creating a regime in which regularization is necessary for avoiding interpolation. We view each sample as belonging to a cluster: for the settings with a total sample size of $N = 180$, we viewed the samples as belonging to 30 different clusters, each of size 6, whereas for the settings with a total sample size of 250, we considered there to be 50 clusters, each of size 5.

After simulating the design matrices $X_i$ and $Z_i$ for each cluster $i$, we generated the response $y_i$ according to Eq (1). In the settings with a sample size of $N = 180$, we chose only 5 (out of $p = 500$, corresponding to a sparsity of $5/500 = 1\%$) of the components of the fixed effect vector $\beta$ to be non-zero and included only a random intercept, but no random slopes ($q = 1$). We refer to the set of indices of the non-zero regression coefficients $\{1 \leq j \leq p : \beta_j \neq 0\}$ as the "active set". In the larger scale simulations in which $N = 250$ and $p = 1000$, we conducted a more extensive exploration of different combinations of fixed effect active set sizes and random effect structures. For the fixed effects, we considered the original active set, as well as the potential impact of doubling its size to match the sparsity of the smaller simulations (10/

**Table 1. Gene expression simulation settings.** Fourteen different settings under which we generated transciptomics data sets. Setting differed in sample size ($N$), number of clusters ($g$), number of fixed effects ($p$), number of random effects ($q$), correlation in auto-regressive covariance matrix ($\rho$), number of non-zero fixed effects (# effects) and random effects covariance ($\Psi_\theta$) structure. For values of the parameters, see Table A in S1 Tables. Setting 2 is identical to setting "H2" in [13] and the parameters used are identical.

|    | $N$ | $g$ | $p$ | $q$ | $\rho$ | # effects | $\Psi_\theta$ structure |
|----|-----|-----|-----|-----|--------|-----------|-------------------------|
| 1  | 180 | 30  | 500 | 1   | 0      | 5         | scalar                  |
| 2  | 180 | 30  | 500 | 1   | 0.6    | 5         | scalar                  |
| 3  | 250 | 50  | 1000| 1   | 0      | 5         | scalar                  |
| 4  | 250 | 50  | 1000| 1   | 0.6    | 5         | scalar                  |
| 5  | 250 | 50  | 1000| 1   | 0      | 10        | scalar                  |
| 6  | 250 | 50  | 1000| 1   | 0.6    | 10        | scalar                  |
| 7  | 250 | 50  | 1000| 3   | 0      | 10        | scalar                  |
| 8  | 250 | 50  | 1000| 3   | 0.6    | 10        | scalar                  |
| 9  | 250 | 50  | 1000| 3   | 0      | 10        | diagonal                |
| 10 | 250 | 50  | 1000| 3   | 0.6    | 10        | diagonal                |
| 11 | 250 | 50  | 1000| 3   | 0      | 10        | unstructured            |
| 12 | 250 | 50  | 1000| 3   | 0.6    | 10        | unstructured            |
| 13 | 250 | 50  | 1000| 5   | 0      | 10        | scalar                  |
| 14 | 250 | 50  | 1000| 5   | 0.6    | 10        | scalar                  |

$1000 = 5/500 = 1\%$). For the random effect structure, we varied the number of random slopes, investigating $q \in \{1, 3, 5\}$; the $q = 1$ case corresponds to a model with only random intercepts, while the other two cases include random slopes for two or four of the predictors, respectively. Furthermore, with $q = 3$, we simulated data under three different choices of the covariance matrix of the random intercept and slopes: a scalar matrix (multiple of the identity), a diagonal matrix, and a general symmetric positive semi-definite matrix. For $q = 5$, we considered only a scalar covariance matrix to manage the complexity of the model.

Table 1 summarizes each of the settings entertained in our gene expression simulations, and Table A in S1 Tables details the values of the parameters $\beta$, $\Psi_\theta$, and $\sigma^2$ in these simulations ($\Psi_\theta$ in table, $\beta$ and $\sigma^2$ in caption). Note that our setting 2 is identical to setting "H2" in [13] (with near identical parameters; our $\sigma^2$ is double that of [13]). However, we go beyond [13] in our extensive exploration of other settings, focusing on the impact of dimensionality, sparsity, and random effect structure. In each of the fourteen settings under investigation, we simulated 100 data sets, consisting of matrices $X$ and $Z$, response vector $y$, and a vector tracking the group membership of each observation.

**2.3.2 Genome-wide association studies (GWAS) simulations.** For the second study, we investigated a small-scale GWAS regression. Whereas a large-scale GWAS might contain hundreds of thousands of SNPs, we anticipated the algorithm, even with the active set modification, only handling problems on the scale of one thousand SNPs without significant computation time. Using the ggmix R package [62], we generated 100 GWAS design matrices of size $N = 250$ by $p = 1000$ with entries representing minor allele counts in a diploid organism (0, 1, or 2). The first $q - 1$ columns of these matrices were assigned random effects. We partitioned the 250 samples from each design matrix into $g = 10$ clusters of size $n_i = 25$, significantly fewer than in the gene expression simulations (settings 3–14 of the gene expression simulations had $g = 50$ clusters, for example). In the GWAS context, clusters may represent distinct, perhaps geographically dispersed, populations. From each simulated design matrix, we used the five random-effect structures considered in the gene expression simulations ($q = 1$, $q = 3$ with scalar, diagonal or unstructured covariance matrix, and $q = 5$ with a scalar covariance

matrix) to generate five different response vectors $y$ according to the mixed-effects model in Eq (1), with the same values of the fixed and random effect parameters as used in the gene expression simulations (Table A in S1 Tables).

**2.3.3 Microbiome simulations.** Finally, we explored the application of high dimensional mixed-effects models fit by CD to microbial operational taxonomic unit (OTU) data. To simulate microbiome data with realistic sparsity levels and covariance structures, we relied on functions in the R packages SPRING and SpiecEasi [63, 64]. At a high level, we generated our count data to mimic the marginal distributions of the 127 OTUs measured in the American Gut Project [65], and we specified different OTU correlation structures in our generated count matrices based on an assumed underlying microbe network. In particular, we considered the six different OTU-network structures provided in the R package SpiecEasi [63]: *band*, *cluster*, *scale free*, *Erdös-Rényi*, *hub*, and *block*. For each network type, we extracted the corresponding inter-OTU covariance structure and then generated 100 OTU count matrices following this covariance structure with sample size (number of rows) 120 and unique OTUs (number of columns) 127. The marginal distribution of the counts of a given OTU was forced to match the empirical cumulative distribution of the OTU counts from the American Gut Project data using the function synthData_from_ecdf in the R package SPRING [64]. Once we had generated these count matrices (100 for each network inter-OTU covariance structure, 600 in total), we transformed them by first adding pseudo counts to the zero entries and then applying a log-ratio transformation, using the last column as the reference. We partitioned the samples into 10 groups of 12 samples each, plausible cluster sizes for microbiome studies, and the first $q - 1$ columns of the log-ratio transformed matrices were assigned random effects. From each design matrix, we used the same five random-effect structures considered in the GWAS simulations to generate five different response vectors according to the mixed-effects model (Eq 1). We included in the generation of the response the additional fixed effect of an additional variable simulated to differ only at the group level (we simulated this variable independently for each design matrix and random-effect structure combination). We retained the same fixed effect coefficients from the previous two settings, with the addition of the coefficient on the group-level variable chosen to be -1. We altered, however, the parameters in the diagonal and unstructured random-effects variance-covariance matrices (Table B in S1 Tables).

## 2.4 Real omics data

In addition to our simulations, we fit high-dimensional mixed-effects model with coordinate descent to published data sets representing real examples of the three omics studies. We now provide brief descriptions of these data sets.

**2.4.1 Bacterial gene expression and riboflavin production.** The riboflavin data set is a popular high throughput transcriptomics data set with a longitudinal design [13]. Specifically, the data set contains repeated measures of the production of riboflavin inside cultures of bacteria (recombinant *Bacillus subtilis*) over a series of generations. These repeated measures allow for the analysis of changes in riboflavin production over time. Thus, observations are clustered within individual bacterial cultures with 28 distinct cultures, each having between 2 to 6 repeated measures at different time points, totaling 111 samples. For each observation, we have measurements of the expression levels of 4,088 bacterial genes and wish to use these to predict the riboflavin production rate. The goal is to identify which genes are most predictive of riboflavin production.

**2.4.2 Mouse GWAS study for body mass index.** This experiment was carried out by [66] to identify genetic signal for complex traits in a population of mice living in $g = 523$ different

cages, which represented the grouping structure for our model. In many mice experiments, cages often contribute significant environmental effects to the phenotypes such as body mass index (BMI), and mice in the same cage tend to be correlated in their phenotype measurements. It is therefore important to account for such cage effects in genetic association studies. Using the high dimensional SNPs as predictors, we estimated a model for the body mass index (BMI) phenotype with cage-specific random intercepts. This approach allows us to account for the variability introduced by different cages. We included as additional unpenalized predictors in the model the age, gender and litter of the mouse (factor with 8 levels), imputing the median age for the 81 mice in which it was missing. This data was analyzed in a similar fashion by the authors of the `BGLR` package in R, using a Bayesian generalized linear model [67] that also included cage-specific random intercepts. Their model, however, did not include age as a predictor. This dataset was also re-analyzed recently in [46], who propose a quasi-likelihood estimation approach to fitting high dimensional mixed-effects models. Their model, however, does not include fixed litter effects. We comment on similarities and divergence between our results and these other analyses in the results.

**2.4.3 Human gut microbiome data across age and geography.** This data, available for download from Qiita (https://qiita.ucsd.edu/) with study ID 850, consists of microbial profiles (based on 16S gene sequencing) of individuals from three different countries ranging in age from infant to 83 years old. The original data set included 14,170 OTUs measured in 528 individuals (315 from the US, 114 from Malawi, and 99 from Venezuela). Of these, 488 individuals had their age recorded in the dataset, which is the variable we model in our analysis (308 from the US, 83 from Malawi, and 97 from Venezuela). Following [25], we reduced the dimensionality of the problem by keeping only OTUs that were present in at least 10% of samples and whose median read count among samples in which they were found was at least 10. This left us with a count matrix of 1,362 OTUs. We added pseudo-counts of 0.5 to this matrix and then applied the center log-ratio transformation. Based on residual diagnostics, we dropped three outlier samples for a final sample size of 485.

In [25], the authors apply a predictive model for age to this data. However, because their model could not accommodate the country-based clustering of samples, they included only the individuals from the USA in their analysis. In contrast, we model the age of individuals from all three countries using our penalized mixed-effects model, including country-unique random intercepts.

## 3 Results

### 3.1 Application to simulated data

We investigated the performance of the estimation procedure described in Section 2.2 in simulations for which we know the true underlying parameters, and we test whether the procedure is robust to the data matrices resembling those found in each of these three omics studies described in Section 1.1. Within each type of omics study, we considered several different settings, varying aspects of the data such as inter-feature correlation structure, number and covariance structure of the random effects, and for the gene expression data, the sample size and dimensionality of the problem, as detailed in Section 2.3.

For each setting in each simulation study, we generated 100 data sets and fit a model to each one in order to obtain Monte Carlo estimates of estimation and variable selection performance under each setting. In each setting, we fit a well-specified model (i.e. we parameterized the covariance matrix for the random effects to include the true data generating random effects covariance matrix), and for each data set, the estimation of the model was done according to the procedure detailed in Section 2.2. Namely, across a grid of values of the regularization

parameters λ, we attempted to minimize Eq (2), with penalty given by either the LASSO or SCAD, through the proposed CD algorithm. We then chose the model that minimized the BIC across all the values of λ. In practice, we found that the solution to which the algorithm converged was quite sensitive to the choice of λ. With λ too low, the model converged to an interpolating solution, whereas setting λ too high resulted in all penalized regression coefficients being forced to zero. The range of λ that resulted in fits that were in between these two extremes could, for some data sets, be quite limited. We provide the grid of λs that we searched over for the gene expression simulations in Table C in S1 Tables; details on the grids for the other two simulation study types, GWAS and microbiome, are contained in the table caption.

We define the "final model fit" for each data set to be the fitted model that minimizes the BIC across the explored grid of λs. In the following plots, we display the distribution of different statistics obtained from this final model fit across the data sets generated under a particular setting. In evaluating the final model fit's performance, we focus primarily on the selection of the correct non-zero coefficients and on the accuracy of estimation for coefficients that are correctly selected. To evaluate variable selection, we view each fitted model as performing a binary classification of the regression coefficients. In this context, we refer to coefficients that have been set to zero in the fitted model as "negatives" and coefficients estimated non-zero as "positives". A true positive is a coefficient correctly estimated non-zero, meaning that its ground-truth value is also non-zero. A false positive is a coefficient incorrectly estimated non-zero. We define the false positive rate (FPR) for a given model fit to be the proportion of ground-truth zero regression coefficients that are mistakenly selected (i.e. estimated non-zero):

$$\mathrm{FPR} = \frac{\#\{j \mid \beta_j = 0 \text{ and } \hat{\beta}_j \neq 0\}}{\{j \mid \beta_j = 0\}}$$

We first discuss the results from simulating gene expression data, then GWAS, and finally microbiome.

**3.1.1 Gene expression.** As mentioned, we considered several different settings in each of the three omics simulation studies. For the gene expression simulations, there were fourteen (14) different settings that we used to generate different gene expression data sets (Table 1). One of the dimensions on which these settings varied was the number of random effects, $q$, that went into generating the response. With $q = 1$, there was only a random intercept in the data generating process, whereas for $q = 3$ and $q = 5$, there were two or four additional random slopes, respectively. In the main text, we focus our attention on the six setting with $q = 3$ random effects (settings 7–12 in Table 1) and display results for all other settings in S1 and S2 Figs.

**Variable Selection.** Each box plot in Fig 1A displays the distribution across data sets of false positive rates of the final model fit upon employing the LASSO (orange) or SCAD (blue) penalty to each of the the data sets simulated under a given setting, and in Fig 1B, the height of each bar indicates the proportion of data sets generated under a given setting for which the model recovered all 10 true non-zero coefficients (i.e. recall was perfect).

In all but three out of the fourteen settings, the median number of false positives across data sets when using the SCAD penalty was zero. Relative to the LASSO, the SCAD-based estimator included false positives less frequently and in smaller number in every setting other than setting 10 and 12 (Fig 1A, bottom right facets). In these settings, it was not the case that the LASSO resulted in good performance. Rather, these were settings in which, for many data sets, every choice of λ led the algorithm to converge either to a solution in which all but the non-penalized coefficients were set to 0 (when λ was chosen above some threshold) or else led the

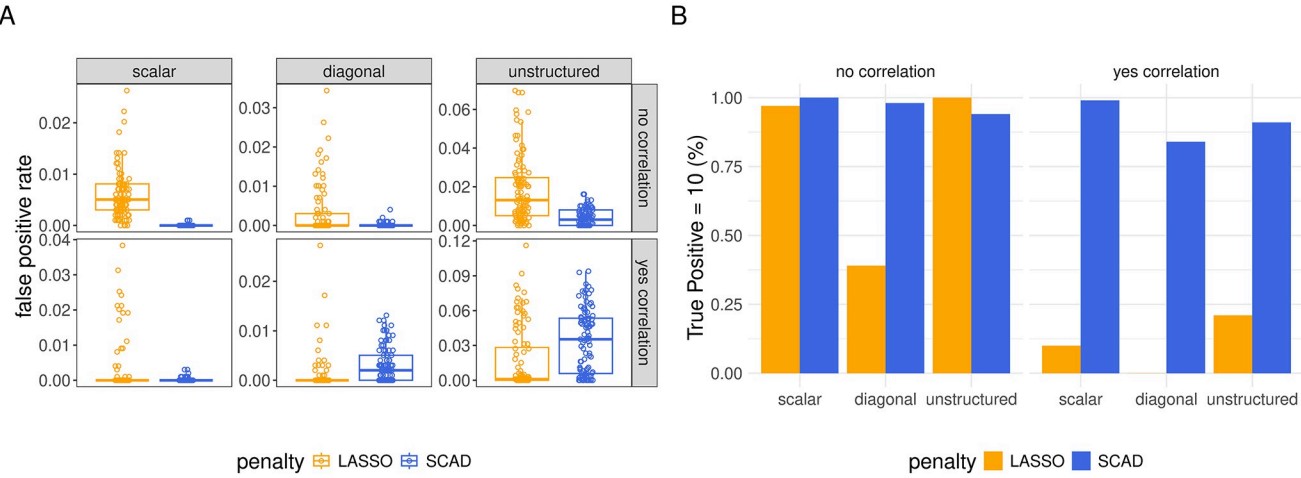

**Fig 1. Variable selection of final model fits in all gene expression simulation settings with $q$ = 3 random effects.** In panel A, the y-axis corresponds to false positive rate. The column faceting identify different random effect covariance ($\Psi_\theta$) structures, while the row faceting indicates the presence of inter-feature correlation: "no correlation" = no correlation between covariates, "yes correlation" = correlation of $\rho = 0.6$ between adjacent covariates in the design matrix (see Section 2.3 for details). In panel B, the height of each bar indicates the proportion of data sets in which the model recovers all 10 true non-zero coefficients. For more complete information about true positive rates in these settings, see Table D in S1 Tables. For the false and true positive rates in all other settings ($q \neq 3$), see S1 Fig.

algorithm to diverge to an interpolating solution with $\sigma^2$ sent to 0 (when λ was below that threshold), when using the LASSO. Since we prefer the former (overly sparse) solution on the basis of BIC, the LASSO-based "final model fit" is not recovering any of the penalized coefficients for these data sets, which means the true positive rate is close to 0 for the LASSO-based estimator in these settings.

Summarizing the variable selection results, the SCAD penalty is to be preferred relative to the LASSO in all settings, as it reliably recovers the non-zero coefficients at the cost of a small number of false positives, and the SCAD is especially appealing in settings with correlated features, since for these settings, the LASSO-based algorithm often cannot be made to converge to a sparse solution without setting all penalized coefficients to 0.

**Estimation.** The LASSO penalty is known to bias its estimates of non-zero parameters towards zero [60]. In our gene expression simulations, we find that this LASSO-penalty-based bias can lead to bias in the estimates of even the coefficients that are not penalized (in the context of mixed-effects models, such unpenalized coefficients are often the coefficients on features for which there are random effects, i.e. columns of $Z_i$).

In Fig 2, we show estimates of all the non-zero regression coefficients from the simulation setting which had inter-feature correlation and $q$ = 3 random effects with unstructured covariance (setting 12 in Table 1). In Fig 3, we display the estimates of one of the penalized regression coefficients, $\beta_4$, and one of the unpenalized regression coefficients, $\beta_3$ across all gene expression data sets simulated with $q$ = 3 random effect (settings 7–12 in Table 1) (analogous plots for all other settings are shown in S2 Fig). The LASSO biases estimates of all penalized coefficients towards 0, and this bias on penalized coefficients can contaminate estimates of the non-penalized coefficients. Specifically, in the settings with inter-feature correlation, the LASSO-based estimates of the non-penalized $\beta_3$ form two clusters, neither of which are centered on the true parameter $\beta_3 = 4$, but which are biased to varying degrees. This bias results from the penalization applied to $\beta_4$, as $\beta_3$ and $\beta_4$ are the coefficients on adjacent columns in the design matrix, which have correlation $\rho = .6$ in these settings. The upward bias in the estimator for $\beta_3$

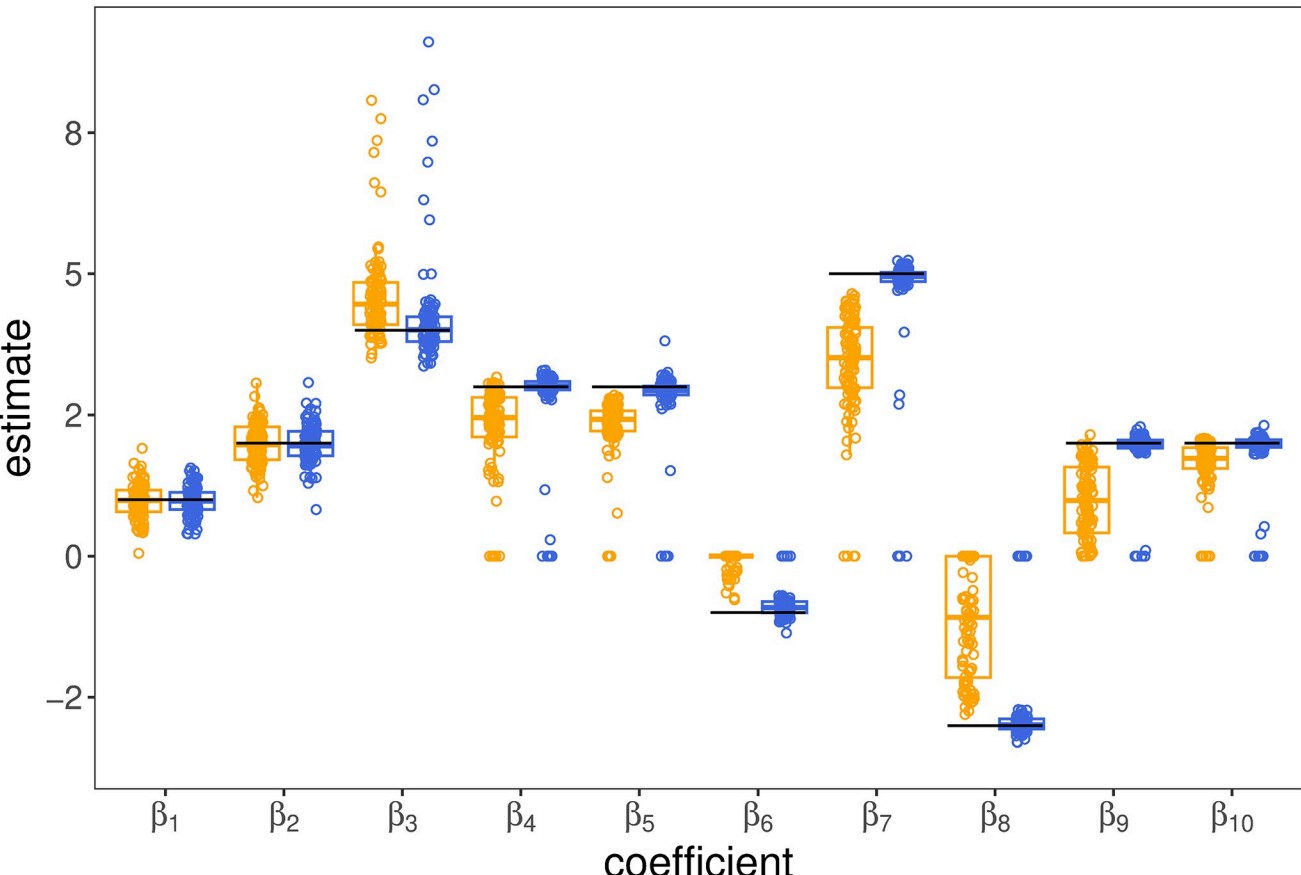

**Fig 2. Estimates of all non-zero coefficients in the gene expression simulation setting with inter-feature correlation and $q$ = 3 random effects with an unstructured covariance matrix (setting 12 from Table 1).** $\beta_1$, $\beta_2$, and $\beta_3$ were unpenalized because they were coefficients on variables with random effects ($\beta_1$ is the intercept). All other coefficient estimates were penalized.

is compensating for the downward bias in the estimate of $\beta_4$, with the required compensation largest when $\beta_4$ is completely excluded from the selected variables, resulting in the observed clustering in the estimates of $\beta_3$. This is additionally visualized in S3 Fig, a scatter plot of the LASSO-based estimates of these two coefficients in the presence of inter-feature correlation. Because the estimation of $\beta_4$ under the SCAD penalty is much more accurate—it is correctly included in the active set in the vast majority of cases and estimated without bias when it is—its bias in the estimation of $\beta_3$ (in the settings with inter-feature correlation) is much less acute than the LASSO's. In general, the SCAD estimates were roughly unbiased, conditional on the model identifying the correct variables, in all settings (blue box plots are centered on the true parameter value which is represented by a solid black line in Fig 3), and under both the LASSO and SCAD penalty, the variability in the estimates of penalized effects was smaller than that of the estimates of unpenalized coefficients.

In most applications, the estimation of variance components is of secondary importance relative to the estimation of fixed effect parameters. S4 Fig shows our estimates of elements of scalar, diagonal, and unstructured random effect covariance matrices, respectively, in the settings with 10 non-zero regression coefficients. Diagonal entries of these random effects

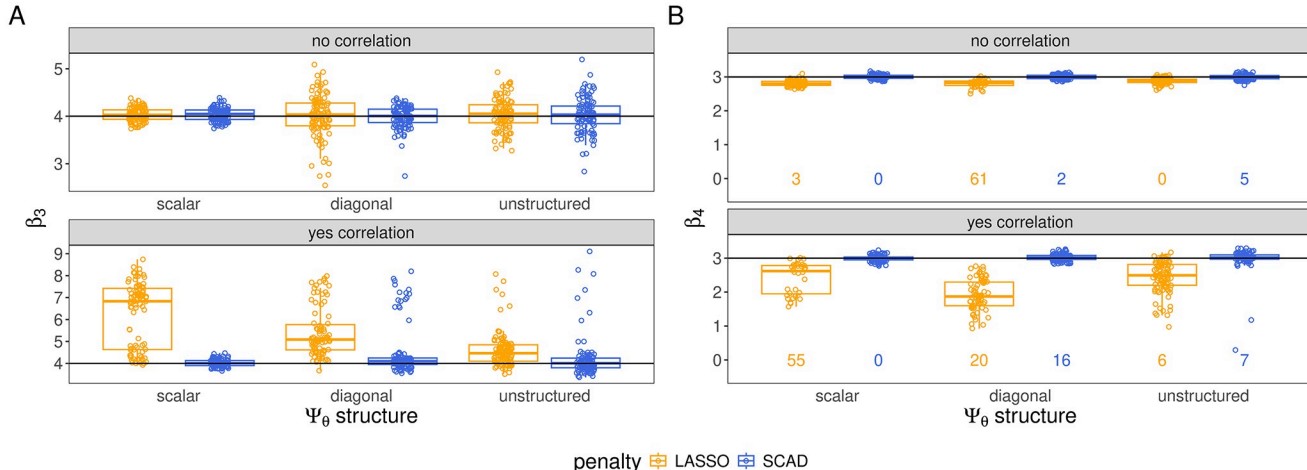

**Fig 3. Estimation of $\beta_3$ (A) and $\beta_4$ (B) from final model fits in all gene expression simulation settings with $q = 3$ random effects.** Panel A shows the distributions of estimates of an unpenalized regression coefficient, $\beta_3$, and panel B shows the distributions of estimates of a penalized coefficient, $\beta_4$. Numbers at the bottom of the plotting windows in panel B indicate the number of data sets for which the coefficient was incorrectly estimated 0. The box plots represent the distribution of the estimator across all other data sets (i.e. all data sets in which the estimate was non-zero). The true parameter value is represented by a solid black line.

covariance matrices were estimated zero when the estimated fixed effect vector included many false positives, as these additional non-zero coefficients proved to be an alternative means to fit the variability driven by the random effects. The LASSO estimates included false positives more frequently, and in these models with many false positives, the random effect variance components were frequently set to zero.

**3.1.2 GWAS.** For the GWAS simulations, we fit a model with SCAD penalty and a model with LASSO penalty to each combination of design matrix and response vector.

**Variable Selection.** We illustrate the performance of our estimators at the task of variable selection in the GWAS setting for the data sets with $q = 3$ random effects in Fig 4 (results in simulations with $q = 1$, 5 are shown in S5 Fig). Both SCAD and LASSO estimators were able to recover the 10 true non-zeros with similar consistency, only occasionally missing one or more of them (Fig 4B). Figs 4A and S5A illustrate, however, that the LASSO penalty led to more false positives than the SCAD penalty in each setting.

**Estimation.** We display the distribution of the estimates of one unpenalized regression coefficient, $\beta_3$, and one penalized regression coefficient, $\beta_8$, from the GWAS simulations with $q = 3$ in S6 Fig (estimates for settings with $q = 1$ or 5 in S7 Fig). In the GWAS simulations, because the features (counts of minor alleles at different locations in the genome) are uncorrelated, we observed no bias in the estimation of non-penalized coefficients under the LASSO (and as usual, the SCAD-based estimators were also unbiased). On the other hand, the bias towards zero in estimates of penalized coefficients under the LASSO penalty remained a problem in the GWAS simulations (S6B Fig).

In terms of the variance components, we observe that the estimates were less accurate than the estimates in the gene expression simulations (S8 Fig). Of course, we can attribute this to the reduction in the number of clusters from $g = 50$ in the gene expression settings 3–14 from Table 1 to $g = 10$ in the GWAS simulations. In general and as previously mentioned, variance components are typically of secondary interest to the fixed effect estimates in omics studies. Nonetheless, it is worth noting that their estimation improves with the number of clusters, as one would expect from experience with low dimensional mixed-effects models.

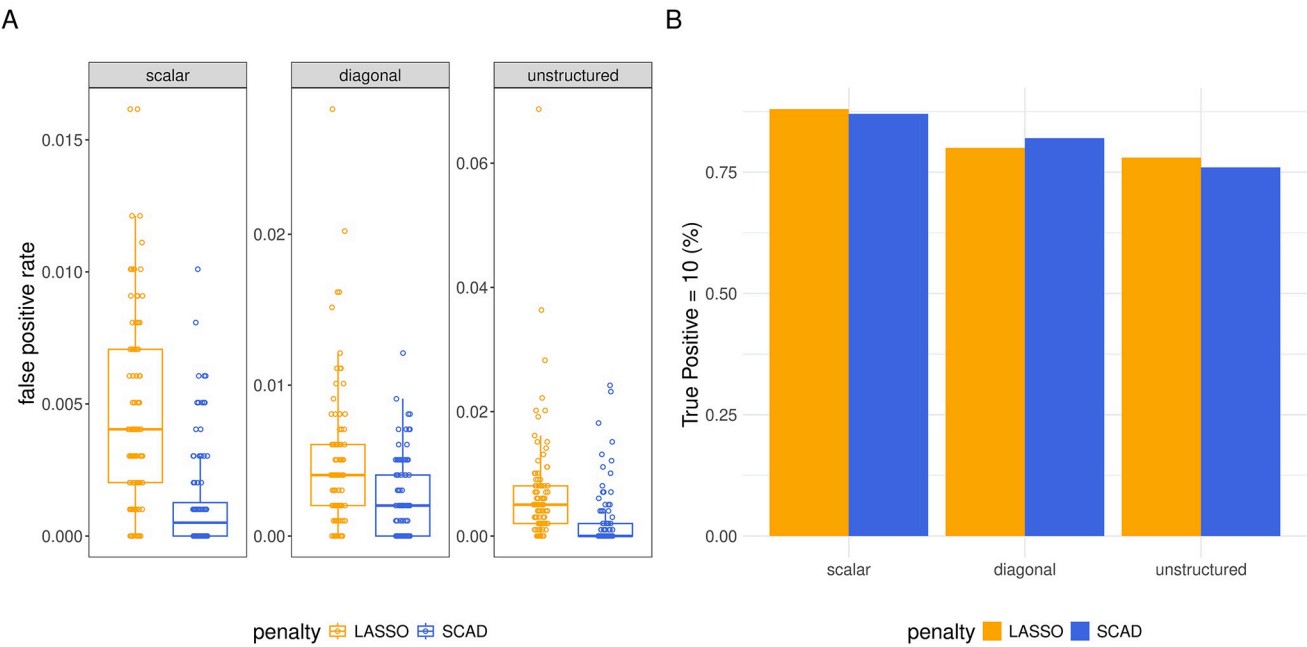

**Fig 4. Variable selection for GWAS data.** Performance at task of variable selection on GWAS data simulated with $q = 3$ random effects. In panel A, Y-axis corresponds to false positive rate and the facets identify different random-effect covariance matrix structures ($\Psi_\theta$). In panel B, the height of each bars indicates the proportion of simulated data sets in which the model identifies all 10 true non-zero coefficients. Colors again differentiate the penalty used to fit the model. For complete results across different ranges of true positive rates, refer to Table E in S1 Tables.

**3.1.3 Microbiome data.** Finally, we describe our results in the microbiome setting. Recall that for these simulations, we simulated OTU count matrices under six different assumed latent OTU network structures, as we were interested in the downstream impact of the count matrices' correlation structure on model performance.

We fit models to each microbiome data set, consisting of a log-ratio transformed microbiome design matrix and response vector. Having observed the superior performance of SCAD relative to LASSO penalization in the other two settings, we focused on only the SCAD penalty in the microbiome setting. We refrained from penalizing the coefficient on the variable measured at the group level (and as usual, did not apply a penalty to coefficients that had corresponding random effects). With the addition of the coefficient on this group-level variable, there were thus 11 non-zero ground-truth regression coefficients for these simulations, which we hoped to recover and accurately estimate.

**Variable Selection.** Since the "band" network structure produced design matrices that proved to be the most challenging to the model, whereas the "scale-free" structure led to the best results, we display model performance from these two network structures in the main text to communicate an accurate sense for the range of results. Fig 5 shows the performance of the estimation procedure at the task of variable selection for these OTU structures for simulations with $q = 3$ random effects.

We recovered all eleven regression coefficients in almost every case when design matrices were generated from a scale-free OTU network, and when the design matrices was generated from a band OTU network but the random effect covariance matrix was scalar. When the design matrices were generated from a band OTU network and the random effect covariance matrix was more complex (diagonal or unstructured), the recall of the model suffered, and in particular, we frequently converged to a solution with all penalized coefficients set to zero

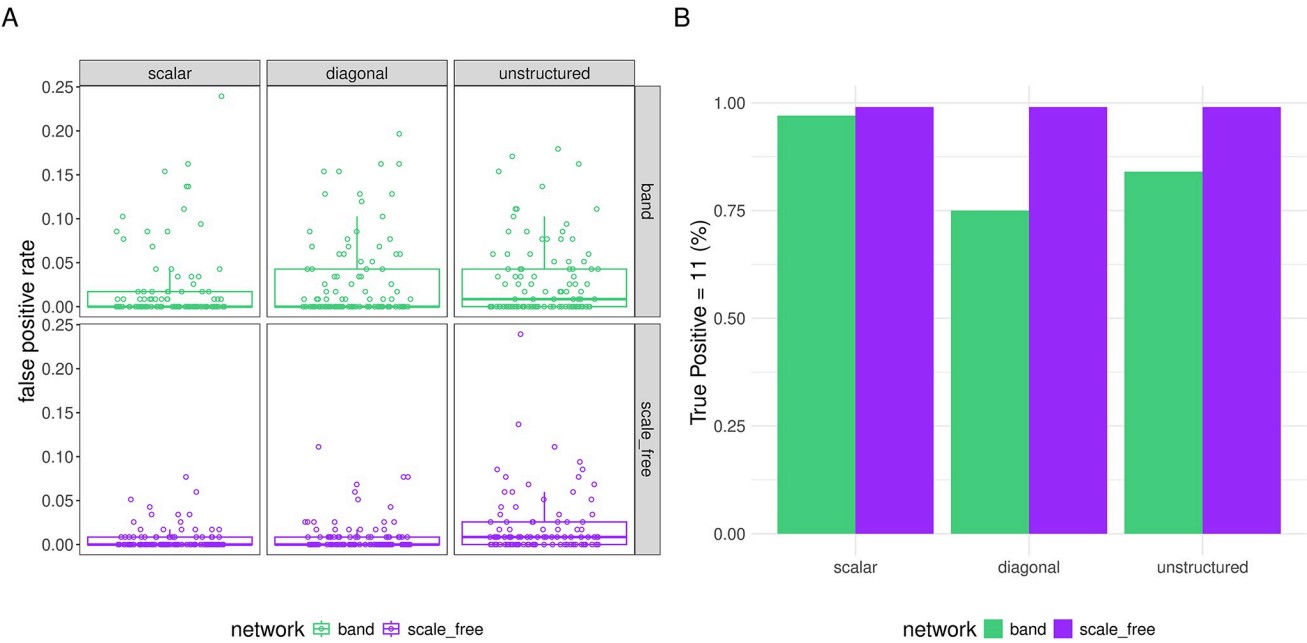

**Fig 5. Variable selection for microbiome data.** Performance at task of variable selection on microbiome data simulated under a "scale-free" or "band" OTU network structure and with response generated with $q = 3$ random effects. In panel A, the columns identify different random effect covariance ($\Psi_\theta$) structures, while the row faceting (and color) indicates the OTU correlation-structure. The Y-axis corresponds to false positive rate. In panel B, the height of each bar indicates the proportion of simulated data sets in which the model identifies all 11 true non-zero coefficients. For complete results across different ranges of true positive rates, refer to Table F in S1 Tables.

(resulting in only four true positives, corresponding to the non-penalized coefficients) (see Table F in S1 Tables). The false positive rates were comparable across settings, with slightly higher false positive rates in the settings that had worse recall, as the model compensated for setting true effects to zero.

The variable selection results for the scale-free and band network structures with $q = 1$ and $q = 5$ are displayed in S9 Fig. We see that the model struggled when there were $q = 5$ random effects and a band design matrix, frequently setting all penalized coefficients set to zero. In the other four OTU network structures, the variable selections results were on par with the results on the scale-free results or slightly worse (S10 Fig). Notably, the algorithm did not converge as frequently to a solution with all penalized coefficients set to zero under these other four OTU-OTU covariances as it did with the band OTU-OTU covariance.

**Estimation.** The fixed effect coefficient on the variable measured at the group level was estimated without bias in the OTU simulations (shown for $q = 3$, band and scale-free network structures in S11 Fig, and other settings in S12 and S13 Figs). Since we used a SCAD penalty, we also estimated all coefficients of covariates measured at the individual level without bias, as usual. There was more variance in the estimates of the coefficient on the group-level variable, which is not surprising, given that there are far fewer groups than individuals. The data sets that led to extreme outliers in our estimates of this coefficient were the ones in which the estimate of penalized coefficients were mistakenly set to zero, frequently data sets with a band OTU covariance structure and three or five random effects.

**3.1.4 Comparing results across omics simulations.** To better visualize and compare the performance of the HigDimMM approach across different omics studies, Fig 6 shows the standard deviations of non-zero estimates for penalized coefficients across various simulated data

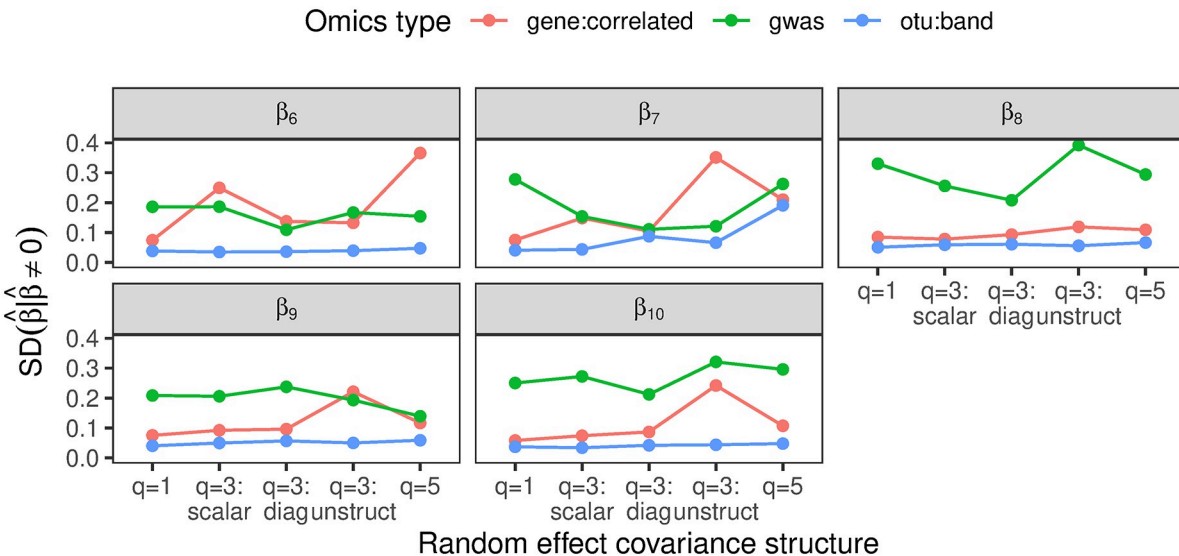

**Fig 6. The standard deviations across simulated data sets of non-zero estimates of penalized coefficients under different omics studies.** For the gene expression study, we show results for the settings with design matrices with non-zero inter-feature correlation and for the microbiome study, for the settings with design matrices with band OTU-network based correlation structure.

sets. The benchmarking was performed under different settings for gene expression, genome-wide association, and microbiome data. For the gene expression study, we focused on design matrices with non-zero inter-feature correlations, while for the microbiome study, the matrices were based on a band OTU-network correlation structure. The HigDimMM with SCAD penalty performs better on OTU band data compared to GWAS data, as indicated by the lower standard deviations for OTU data. The higher variability in GWAS data suggests that the model is less effective in stabilizing coefficient estimates in this setting, potentially due to the complexity of the underlying genetic data structure.

## 3.2 Application to real data

We now turn to applications of the high dimensional mixed-effects model on real omics datasets.

**3.2.1 Bacterial gene expression and riboflavin production.** The riboflavin dataset is a longitudinal dataset of bacterial gene expression that was originally made available with [7] and was analyzed by Schelldorfer et al. in [13] (see Section 2.4.1 for further details). Following the strategy proposed in [13] for selecting random effects, we assign random effects to genes *YFJD* and *YTOI*. We fit a model that included random slopes for these two genes (and in contrast to previous models, no random intercept) with a diagonal covariance and with $\lambda$ set to 45 based on the BIC. The fitted model selects the 17 genes listed in Table 2 as potentially impacting the riboflavin production rate. We compared the results of fitting the model with our Julia implementation to the results of fitting with the existing R implementations of the algorithm under the LASSO and SCAD penalties, which we denote *lmmlasso* [13] and *lmmSCAD* [14], respectively. When fitting with the R implementations, we similarly assigned genes *YFJD* and *YTOI* random effects, excluded a random intercept, and specified a diagonal covariance matrix. The *lmmlasso* model identified 28 important genes, of which 3 (*TUAH*, *YXLD*, *YDDK*) were found to be common with our gene list. For *lmmSCAD*, the analysis yielded 5 intersecting

**Table 2. Non-zero estimated gene effects in riboflavin data.** Genes with non-zero estimated effect on riboflavin production using our HighDimMM model.

| Gene | Estimated-effect | Gene | Estimated-effect |
|------|------------------|------|------------------|
| YURQ | 1.427 | YCGP | -0.103 |
| YFKE | 1.073 | YDBM | -0.103 |
| YTOI | 1.019 | YTXM | -0.115 |
| YFJD | 0.721 | UVRB | -0.176 |
| YLMA | -0.003 | YNEI | -0.235 |
| YUSY | -0.005 | LYSC | -0.383 |
| METK | -0.025 | YXLD | -0.526 |
| YUBB | -0.055 | TUAH | -1.343 |
| YDDK | -0.1 | | |

genes (*LYSC, TUAH, YURQ, YXLD, YDDK*). The complete gene lists are provided in Table G in S1 Tables, with intersecting genes highlighted in bold.

**3.2.2 Mouse GWAS study for body mass index.** We next applied the high dimensional mixed-effects model to data from a GWAS study in mice with 10,346 polymorphic markers measured in 1,814 individuals (see Section 2.4.2 for dataset details). We fit high dimensional mixed-effects models with the SCAD penalty to the mouse data set, searching over three different values of the regularization hyperparameter λ: 150, 190,and 200. Because of the significantly greater scale of this data set ($N = 1, 814, p = 10, 356$), we specified a convergence tolerance that allowed the algorithm to converge in fewer iterations. For example, with λ = 150, the algorithm converged in only four iterations (only two of which updated all 10,356 parameters because of the active set approach), lasting just under 27 minutes on an Apple M1 processor.

As in all cases, increasing λ resulted in fewer selected loci, and the estimates from all three are shown in Fig 7. The fit that minimized the BIC was obtained with the largest penalty and included only 7 SNPs. The predictive accuracy of this final model was on par with the Bayesian model from [67], as the estimated error variances of the two models were $\hat{\sigma}^2_{\mathrm{BGLR}} = 0.52$ and $\hat{\sigma}^2_{\mathrm{HighDimMM}} = 0.57$, respectively. Because the Bayesian model does not apply a sparsity-inducing penalty, however, it includes all SNPs in the model and is therefore less interpretable than our model fit with the SCAD penalty. We validated the features selected by our model by checking that all 7 SNPs with estimated non-zero coefficients from our model are among those with the top 20 largest coefficient magnitudes in the Bayesian model. Based on Figs 7 and S14, our model is able to reduce the error variance as much as the Bayesian model while using far fewer features by 1) assigning larger effect sizes (coefficient magnitudes) to the selected features and 2) making greater use of the cage effects, as seen by comparing the observed versus fitted plots with and without the inclusion of the random intercepts in S14 Fig.

We additionally compared our results to the analysis of the same dataset performed in [46]. Their model selects 13 SNPs as associated with BMI at a false discovery rate (FDR) of 0.05, in addition to an effect of gender. Of these 13 SNPs, 2 are also selected in our model fit with λ = 150, but none are selected in our model fit with λ = 190 and λ = 200. The two SNPs that overlap with our model fit with λ = 150, which selected 25 SNPs in total, are *rs3023058* and *rs6185805*. These two SNPs are located on the genes *Srrm3* and *Mtcl1*, respectively.

As previously mentioned, we differ slightly from the model specification in [46] in our choice to include mouse litter, a categorical variable that is shared by all mice in the same cage, as a predictor in the model. Our model produced (unpenalized) estimates of 0.4 and 0.5

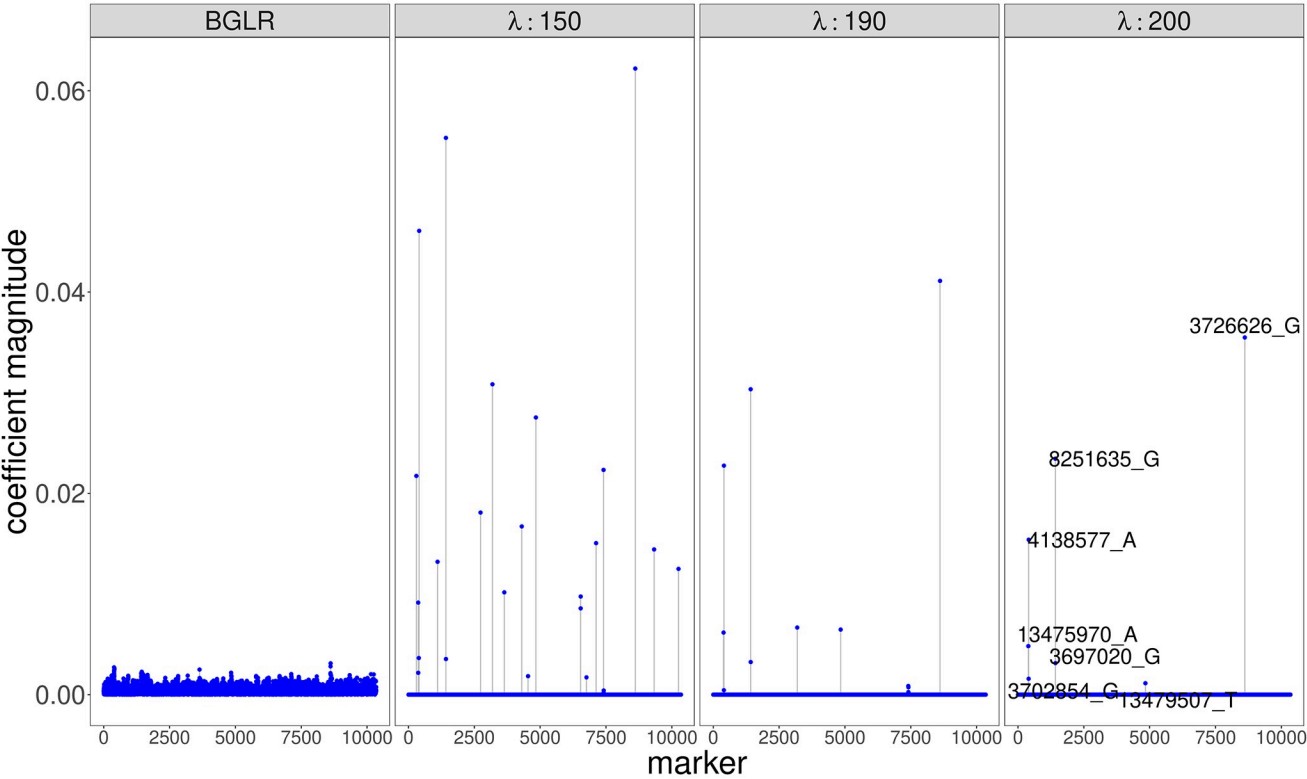

**Fig 7. Absolute values of coefficient estimates from HighDimMM using a SCAD penalty with different values of λ and from a Bayesian model fit with BGLR which does not use a sparsity-inducing penalty and thus includes all SNPs in the model.** The value of λ is given in the strips above the plots–the plot on the far right is from BGLR. For our model, the fit that minimized the BIC was obtained with λ = 200 and included only 7 SNPs, which are labelled in the plot.

(which are substantial effect sizes given that the largest SNP coefficient is an order of magnitude lower at 0.04) for the coefficients on litters 7 and 8 (and negligible coefficients for all other litters). However, given that these were the smallest litters with sample sizes of only 14 and 4, these effects are not necessarily statistically significant, and the inclusion of litter as a predictor likely does not explain the large discrepancy in identified SNPs by our approach and that of [46]. More likely, the lack of overlap in selected SNPs points to a fundamental differences between the likelihood-based approach to fitting the model and the quasi-likelihood approach proposed in [46].

**3.2.3 Human gut microbiome data across age and geography.**   Finally, we applied our model to a study of the diversity of microbial compositions across age and geography [68] (refer to Section 2.4.3 for details on this dataset and its pre-processing). We fit our model to the logarithm of age and included a quadratic term for the OTU with the largest regression coefficient, *Clostridium hiranonis*. Selecting λ = 100, our algorithm converges to a sparse solution with only 9/1362 non-zero regression coefficients. In the context of the data, we identify these 9 OTUs as associated with age (Table H in S1 Tables). The country-specific random intercept variance is estimated as 0.10, and the mean squared error (estimated error variance) is 0.90, representing a reduction of 71% in variance from that of the original response, the logarithm of age.

**3.2.4 Runtimes.**   Table 3 displays the runtime of our algorithm on each of the real omics data sets. Due to its larger sample size and number of features, the runtime was significantly

**Table 3. Runtime of fitting algorithm (for single λ) on each of three real omics data sets.** All computations were performed on an Apple M1 processor.

| Data set | $N$ | $p$ | Convergence tolerance | No. iterations | Runtime (s) |
|---|---|---|---|---|---|
| Riboflavin gene expresion | 111 | 4,088 | 1e-4 | 36 | 121 |
| Mouse GWAS | 1,814 | 10,346 | 1e-3 | 4 | 1,620 |
| Age microbiome | 485 | 1,363 | 1e-4 | 29 | 25 |

longer for the GWAS data set. To facilitate the fitting of the algorithm to this data sets, we increased the convergence tolerance. With this higher convergence tolerance, the algorithm converged in only four iterations in just under 27 minutes.

## 4 Discussion

In this work, we have studied the performance of a CD algorithm for fitting high dimensional mixed-effect models, focusing on three canonical data types in modern biology. Previous studies of the high dimensional mixed-effects model have focused on proving properties about a theoretical global or local maxima of the penalized likelihood. In practice, however, estimates must be obtained by an iterative algorithm that lacks guarantees of convergence to even a local minimum of the penalized likelihood objective function. We have shown that the proposed algorithm with the SCAD penalty in fact modifies the objective function it minimizes at each coordinate update and, therefore, does not converge to even a stationary point of the original objective function. We highlight here that despite or perhaps because the updates in the SCAD-based algorithm do not minimize and indeed often do not even decrease the original penalized likelihood objective function, its convergence behavior is more stable than the LASSO-based algorithm, and when both versions converge to sparse solutions, the particular parameter estimates that are obtained with the SCAD-based algorithm are superior in terms of feature selection and estimation accuracy to those obtained with the LASSO, across simulated omics designs. We have implemented the descent algorithms for the SCAD penalty (as well as the LASSO) in a Julia package, correcting mistakes found in previous implementations, which have either failed to implement the correct update of the penalized coefficients under the SCAD penalty [18] or failed to correctly implement the active set strategy, resulting in an algorithm in which any regression coefficient that is set to zero at a given iteration is never further updated [14]. We hope that our implementation proves useful to biologists working across genomics who wish to fit these models to analyze their clustered, high dimensional data.

At the same time, several limitations of these models have become clear to us over the course of our simulation studies. First, the fitting of these models requires the choice of the regularization hyperparameter λ. Choosing a λ that is too small results in a model that includes many predictors and interpolates the data; meanwhile, a λ that is too large results in a model in which all penalized coefficients are set to zero. While this is a characteristic of all penalized likelihood estimation strategies, it poses a particular challenge for models in which fitting the model for a particular choice of λ is relatively computationally expensive, such as CD algorithms (even with the active set strategy), since the entire model selection process requires repeating this computationally expensive procedure across a grid of λs. For this reason, a *pathwise coordinate descent* approach, in which the algorithm for each λ is initialized with the solution for the previous λ in the path, has been a popular approach to efficiently fit LASSO models without random effects [69]. It is worth exploring whether this approach could be extended to the fitting of high dimensional models with random effects, using either the LASSO or SCAD penalty.

The second major limitation of this approach to analyzing high dimensional, clustered data is the current lack of tools for assessing the degree of certainty associated with the selected features. Buhlmann et al. [7] present a variety of strategies for quantifying feature selection uncertainty in high dimensional biological studies. Although these strategies do not explicitly account for random effects, many of them may still prove useful in the mixed-effects context. One complication is that many of these strategies rely on some form of bootstrapping and in a clustered data context, bootstrapping is less straightforward. Nonetheless, in future work, we hope to adapt some of these strategies in order to quantify the uncertainty and operating characteristics of the features selected by the penalized mixed-effects model. Post-selection inference for mixed-effects models is an area of ongoing research.

Finally, we note that the model and likelihood presented in this paper are that of a continuous, Gaussian response, and we have not provided any advice for biologists working with other types of responses, such as binary or count data. There are existing algorithms and methods that can be used when the distribution of the response is a member of an exponential family—that is, for fitting generalized linear mixed models in the high dimensional setting—that we have not studied in the present work [29, 70, 71]. It is important to have an understanding of the operating characteristics of these methods—in particular, the reliability of their selection of predictors—and a simulation study similar in scope to the one we have conducted in this paper, with dataset examples drawn from the biological study types we have identified but with differently distributed responses, would be valuable towards this end.

## Supporting information

**S1 Appendix. Details on coordinate descent algorithm.**
(PDF)

**S1 Tables.** Table A. **Random effect covariance matrices for the GWAS and gene expression simulations with $q$ = 3 random effects**. For simulations with 1 or 5 random effects, we specified a scalar covariance matrices, and the same scalar was chosen as the $q$ = 3 case, i.e. 0.56. For fixed effect parameters, the non-zero components of the vector of fixed effect coefficients $\beta$ was (1, 2, 4, 3, 3) in the settings with five non-zero effects and (1, 2, 4, 3, 3, −1, 5, −3, 2, 2) in the settings with ten. All other components were zero. All non-zero components appeared at the start of the vector in the order presented. The error variance $\sigma^2$ was chosen to be 0.5 across all gene expression simulations. Table B. **Random effect covariance matrices for the microbiome simulations with $q$ = 3 random effects.** For simulations with 1 or 5 random effects, we specified a scalar covariance matrices, and the same scalar was chosen as the $q$ = 3 case, i.e. 0.56. The error variance and non-zero components of $\beta$ were the same as in the gene expression and GWAS simulations (see caption in Table SA), with the exception that we added one additional non-zero component in $\beta$: a coefficient of −1 on the variable measured at the group level. Table C. **Grid Search Parameters and Convergence Results for Simulation Settings.** We used grid search to select $\lambda$ for each simulated data set, and we searched over different grids depending on the simulation setting and the penalty being applied, as detailed in this table for the gene expression simulations. For the GWAS simulated data, we search over the interval 10 to 100 with increments of 1, and for the microbiome simulated data, we search over the same interval but with increments of 5 in all settings and penalties. These grids, as well as the convergence tolerance hyperparameter, were intentionally chosen so as to avoid convergence issues, since for each setting, a slightly different range of $\lambda$ is necessary to avoid an interpolating solution. Despite these efforts, there were certain gene expression simulation settings that proved especially difficult to fit with a LASSO penalty. In each of these difficult settings, there was at least one data set for which the coordinate gradient descent with a LASSO penalty

converged to an interpolating solution for all of the λs that we searched over. These were setting 5 (one data set proved problematic), setting 6 (one data set proved problematic), setting 7 (three data sets proved problematic), and, most difficult of all, settings 8 and 10 (thirteen data sets proved problematic in each). Thus, while in general, the box plots in the main text show distributions over one hundred data sets each, for these particular settings with the LASSO penalty, the distributions depicted are over slightly fewer than one hundred data sets. When adopting the SCAD penalty, we were able to find at least one non-interpolating solution from the searched-over grid of λs for each data set (from each setting), so all boxplots depicting SCAD results are displaying a distribution over all one hundred data sets. Further, in the other two simulation domains (GWAS and microbiome), we were able to find at least one non-interpolating solution from the searched-over grid of λs in all settings under each attempted penalty. Table D. **The results of true positive gene expression prediction under various conditions.** The columns indicate the type of penalty used (LASSO or SCAD), whether predictor variables are correlated ("Yes" or "No"), the covariance structure (diagonal, scalar, or unstructured), and the percentage of true positive among a total of 100 simulated data across different ranges of values. Table E. **The results of true positive GWAS prediction under various conditions.** The columns indicate the type of penalty used (LASSO or SCAD) and the covariance structure (diagonal, scalar, or unstructured), and the percentage of true positive among a total of 100 simulated data across different ranges of values. Table F. **The results of true positive microbiome prediction under various conditions.** The columns indicate the structure of the network (band or scale_free) and the covariance structure (diagonal, scalar, or unstructured), along with the percentage of true positive predictions among a total of 100 simulated data across different ranges of values. Table G. **List of impactful genes identified when fitting model to Riboflvain data using our implementation (HighDimMixedModels) versus R implementations (lmmlasso and lmmscad).** The intersection of genes between the implementations is highlighted in bold. Table H. **Taxonomic classification of OTUs selected by HighDimMM and their estimated regression coefficients.** *Clostridium hiranonis* was also assigned a quadratic term, but we show here only the coefficient on the linear term. The OTUs are ordered by the magnitude of their coefficient.
(PDF)

**S1 Fig. Variable selection for gene expression data.** Performance at task of variable selection on gene expression data from settings 1–4 (panels A and B) and settings 5–6, 13–14 (panels C and D) (see Table 1 in the main text). Along the x-axis are different features of the simulation (dimension $p$ for A and B and number of random effects $q$ for C and D), and the facets are according to the presence of inter-feature correlation. In A and C, the false positive rate is plotted. In B and D, the number of true positives is plotted.
(EPS)

**S2 Fig. Effect estimation for gene expression data.** Performance at task of estimating non-zero regression coefficients on gene expression data simulated under settings 1–4 (panels A and B) and settings 5–6, 13–14 (panels C and D). The coefficient being estimated in panels A and C is $\beta_3$; in panels B and D, the coefficient being estimated is $\beta_4$. These coefficients were penalized for all the settings depicted in panel A and B, but were not penalized when $q = 5$ in panels C and D. Numbers at the bottom of the plotting windows of panels C and D indicate the count of data sets for which the coefficient was incorrectly estimated 0 (there were no 0 estimates in panels A and B), and the box plots represent the distribution of only the non-zero estimates. The true parameter value is represented by a solid black line.
(EPS)

**S3 Fig. Compensation in LASSO-based estimation in presence of inter-feature correlation.** Scatter plot of $\hat{\beta}_3$ versus $\hat{\beta}_4$ in the LASSO-based estimator in settings with $q = 3$ and inter-feature correlation. $\beta_4$ is a penalized coefficient, $\beta_3$ is not; as the estimate of $\beta_4$ is pulled down by the LASSO penalty, the estimate of $\beta_3$ goes up to compensate, with the most extreme compensation occurring when $\beta_4$ is set to 0. The true values of $\beta_3$ and $\beta_4$ are 4 and 3, respectively, depicted by the black horizontal and vertical lines.
(EPS)

**S4 Fig. Variance component estimation for gene expression data.** Estimates of random effects covariance matrices across the ten settings that had ten non-zero regression coefficients (settings 5–14, see Table 1 in main text). The true parameter value is represented by a solid black line. A shows estimates of the single parameter in a scalar matrix, with x axis indicating settings with one (settings 5–6), three (settings 7–8), or five random effects (settings 13–14); B shows estimates of the 3 parameters in a diagonal random effects covariance matrix (settings 9–10); and C shows estimates of the 9 parameters in an unstructured random effects covariance matrix, only 6 of which are unique due to symmetry (settings 11 and 12).
(EPS)

**S5 Fig. Variable selection for GWAS data with $q = 1$ or 5.** Performance at task of variable selection on GWAS data simulated with $q = 1$ or 5 random effects. Y-axis corresponds to false positive rate in panel A and true positives count in panel B. X-axis corresponds to random effects covariance structure ($\Psi_\theta$). The numbers appearing at the top of B are the counts of data sets among a total of 100 in which all 10 non-zero coefficients were recovered.
(EPS)

**S6 Fig. Effect estimation for GWAS data with $q = 3$.** Distributions of estimates of an unpenalized regression coefficient, $\beta_3$, (A) and of a penalized regression coefficient, $\beta_8$, (B) across all gene expression data simulated with $q = 3$ random effects. The true parameter value is represented by a solid black line.
(EPS)

**S7 Fig. Effect estimation for GWAS data with $q = 1$ or 5.** Distributions of estimates of $\beta_3$ and $\beta_8$ across all gene expression data simulated with $q = 1$ or 5 random effects. In panel A, the coefficient $\beta_3$ was unpenalized for settings with 5 random effects (boxplots on the right), but penalized for settings with only 1 random effect (boxplots on the left). In panel B, the coefficient $\beta_8$ was penalized in both settings.
(EPS)

**S8 Fig. Variance component estimation for GWAS data.** A) shows estimates of the single parameter in a scalar matrix, with the x-axis indicating whether the setting had $q = 1$, 3, or 5 random effects; B) shows estimates of the 3 parameters in a diagonal random effects covariance matrix; and C) shows estimates of the 9 parameters in an unstructured random effects covariance matrix, only 6 of which are unique due to symmetry. The true parameter value is represented by a solid black line in each case.
(EPS)

**S9 Fig. Variable selection for microbiome data with "scale-free" and "band" structure and $q \in \{1, 5\}$.** Performance at task of variable selection on microbiome data simulated under a "scale-free" or "band" OTU network structure and with response generated with $q = 1$ or 5 random effects. Y-axis corresponds to false positive rate in panel A and true positives count in panel B. X-axis corresponds to random effects covariance structure ($\Psi_\theta$). The numbers appearing at the top of B are the counts of data sets among a total of 100, in which all 11 non-

zero coefficients were recovered.
(EPS)

**S10 Fig. Variable selection for microbiome data with "block", "cluster", "Erdos-Renyi", and "hub" structure for all values of $q$.** Performance at task of variable selection on microbiome data with "block", "cluster", "Erdos-Renyi", and "hub" structure. Y-axis corresponds to false positive rate in panel A and true positives count in panel B. X-axis corresponds to random effects covariance structure ($\Psi_\theta$). The numbers appearing at the top of each of the facets in B are the counts of data sets among a total of 100, in which all 11 non-zero coefficients were recovered.
(EPS)

**S11 Fig. Effect estimation for microbiome data with "scale-free" and "band" structure and $q$ = 3.** Performance at task of effect estimation on microbiome data simulated under a "scale-free" or "band" OTU network structure and with response generated with $q$ = 3 random effects. Distributions of estimates of the unpenalized coefficient on the variable measured at the group-level. The true parameter value is represented by a solid black line. Plot windows are cut off at a maximum $y$ = 0 and minimum of $y$ = −2, so that extreme outliers are not shown.
(EPS)

**S12 Fig. Effect estimation for microbiome data with "scale-free" and "band" structure and $q \in \{1, 5\}$.** Performance at task of effect estimation on microbiome data simulated under a "scale-free" or "band" OTU network structure and with response generated with $q$ = 1 or 5 random effects. Distributions of estimates of the unpenalized coefficient on the variable measured at the group-level. The true parameter value is represented by a solid black line. Plot windows are cut off at a maximum $y$ = 0 and minimum of $y$ = −2, so that extreme outliers are not shown.
(EPS)

**S13 Fig. Effect estimation for microbiome data with "block", "cluster", "Erdos-Renyi", and "hub" structure for all values of $q$.** Performance at task of effect estimation on microbiome data simulated under a "block", "cluster", "Erdos-Renyi", and "hub" structure. Distributions of estimates of the unpenalized coefficient on the variable measured at the group-level. The true parameter value is represented by a solid black line. Plot windows are cut off at a maximum $y$ = 0 and minimum of $y$ = −2, so that extreme outliers are not shown.
(EPS)

**S14 Fig. True versus fitted (standardized) BMI from BGLR model and our model (HighDimMixedModels with SCAD penalty) with and without random effects included.** Points are colored by gender. Blue line shows a perfect fit. In top left, the predictions from BGLR model do not include random effects; in top right, they do. Similarly, in bottom left, the prediction from our model do not include random effects; in bottom right, they do. While the population level predictions from our model are quite poor (bottom left), once the cage-level random effect are incorporated, the predictions from both models are equally accurate.
(EPS)

## Author Contributions

**Conceptualization:** Evan Gorstein, Rosa Aghdam, Claudia Solís-Lemus.

**Data curation:** Evan Gorstein, Rosa Aghdam.

**Formal analysis:** Evan Gorstein, Rosa Aghdam.

**Investigation:** Evan Gorstein, Rosa Aghdam.

**Methodology:** Evan Gorstein, Rosa Aghdam, Claudia Solís-Lemus.

**Project administration:** Claudia Solís-Lemus.

**Resources:** Claudia Solís-Lemus.

**Software:** Evan Gorstein.

**Supervision:** Claudia Solís-Lemus.

**Validation:** Evan Gorstein.

**Visualization:** Evan Gorstein, Rosa Aghdam.

**Writing – original draft:** Evan Gorstein.

**Writing – review & editing:** Evan Gorstein, Rosa Aghdam, Claudia Solís-Lemus.

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
