## [Decision Letter · Decision Letter 0]

26 Aug 2024

Dear Dr. Solis-Lemus,

Thank you very much for submitting your manuscript "HighDimMixedModels.jl: Robust High Dimensional Mixed Models across Omics Data" for consideration at PLOS Computational Biology.

As with all papers reviewed by the journal, your manuscript was reviewed by members of the editorial board and by several independent reviewers. In light of the reviews (below this email), we would like to invite the resubmission of a significantly-revised version that takes into account the reviewers' comments.

We cannot make any decision about publication until we have seen the revised manuscript and your response to the reviewers' comments. Your revised manuscript is also likely to be sent to reviewers for further evaluation.

Sincerely,

Sonika Tyagi

Academic Editor

PLOS Computational Biology

Alison Marsden

Section Editor

PLOS Computational Biology

Reviewer's Responses to Questions

**Comments to the Authors:**

Reviewer #1: The reviewer report has been attached.

Reviewer #2: The authors introduce an algorithmic approach suited to longitudinal (i.e. time-series) and high-dimensional biological data i.e. (where p >> n). The main novelty is the generalisability of the approach to many conventional omics data types. Software is accessible under a MIT licence with comprehensive documentation and worked examples.

Major comments:

1. Benchmarking of the various omics studies is carried out, but addressed individually. It would be useful for the reader to see a combined summary table or figure indicating the performance of the HigDimMixedModels approach against the results/models of each study featured in the manuscript.

Line 39-40

“We provide a table summarising existing methods”

It is possible that I missed it, but it seems that this table is missing from the supplementary material at this time? A direct reference to the table number would also be useful.

Minor comments:

1. Scripts, data, documentation and results are available and well-organised, which streamlines the review process, boosts reproducibility and software use. However, it would be useful for the authors to provide an estimate of compute resource use and time for a given dataset (e.g. those outlined in the manuscript and documentation).

2. The scope of this package and study is focused on analysing single-omics data. However, many studies now incorporate multi-omics data analyses instead of single-omics analyses. If I understand correctly, it appears that this work may complement multi-omics studies by standardising data in a generic way though before multi-omics data integration (eg by using the mixOmics package). Can the authors discuss the potential use of this method for this application?

3. The discussion section implies that the authors intend to refine and expand their work. For the future, it would be useful to add biological examples (e.g. those in the manuscript) to the documentation directly covering multiple use case scenarios. Currently, the documentation has an example of its use on a small non-biological dataset, although this is understandable for testing purposes.

Line 362

Binary or count data is common in biology and is often transformed into a continuous distribution for analysis, for example in the case of RNA-Seq data. The authors may be interested in adding some comments on the potential feasibility of adapting this method for non-continuous data.

**Have the authors made all data and (if applicable) computational code underlying the findings in their manuscript fully available?**

Reviewer #1: Yes

Reviewer #2: Yes

PLOS authors have the option to publish the peer review history of their article (what does this mean?). If published, this will include your full peer review and any attached files.

Reviewer #1: No

Reviewer #2: **Yes: **Tyrone Chen
---

## [Decision Letter · Decision Letter 1]

19 Nov 2024

Dear Dr. Solís-Lemus,

We are pleased to inform you that your manuscript 'HighDimMixedModels.jl: Robust High Dimensional Mixed Models across Omics Data' has been provisionally accepted for publication in PLOS Computational Biology.

Best regards,

Marc R Birtwistle, PhD

Section Editor

PLOS Computational Biology

Alison Marsden

Academic Editor

PLOS Computational Biology

Feilim Mac Gabhann

Editor-in-Chief

PLOS Computational Biology

Jason Papin

Editor-in-Chief

PLOS Computational Biology

Reviewer's Responses to Questions

**Comments to the Authors:**

Reviewer #1: Thank you for addressing the comments.

Reviewer #2: The authors have addressed all comments, congratulations on the article.

**Have the authors made all data and (if applicable) computational code underlying the findings in their manuscript fully available?**

Reviewer #1: Yes

Reviewer #2: Yes

PLOS authors have the option to publish the peer review history of their article (what does this mean?). If published, this will include your full peer review and any attached files.

Reviewer #1: No

Reviewer #2: **Yes: **Tyrone Chen

---

## [Editor Report · Acceptance letter]

1 Jan 2025

PCOMPBIOL-D-24-00770R1 

HighDimMixedModels.jl: Robust High Dimensional Mixed Models across Omics Data

Dear Dr Solís-Lemus,

I am pleased to inform you that your manuscript has been formally accepted for publication in PLOS Computational Biology. Your manuscript is now with our production department and you will be notified of the publication date in due course.

With kind regards,

Jazmin Toth
